# Person-Centered Health Intervention Programs, Provided at Home to Older Adults with Multimorbidity and Their Caregivers: Protocol for a Systematic Review

**DOI:** 10.3390/jpm13010027

**Published:** 2022-12-23

**Authors:** Vânia Nascimento, César Fonseca, Lara Guedes Pinho, Manuel José Lopes

**Affiliations:** 1Nursing Department, University of Évora, 7000-811 Évora, Portugal; 2Comprehensive Health Research Centre (CHRC), University of Évora, 7000-811 Évora, Portugal

**Keywords:** older adults, caregivers, patient-centered care, home care

## Abstract

The WHO has been promoting the paradigm shift in health care towards person-centered care, considering this strategy as fundamental for the personalization of care, but globally, the implementation of person-centered health intervention programs is still in an early stage. Older adults have high morbidity rates, which are often precursors to functional dependence on informal caregivers. Person-centered health intervention programs may answer the vulnerability of older adults and their caregivers, but they are not yet intensively implemented. This systematic literature review aims to identify which person-centered health programs exist in-home settings for this population and show the health gains. Methods: A systematic review of the literature will be conducted in the following databases: PubMed, CINAHL, MedicLatina, Scopus, and MEDLINE. The search strategy will contain the following MesH terms or similar: “older adults”, “caregivers”, “home care”, and “patient-centered care.” Criteria inclusion: Person-centered health intervention programs performed to older adults and their caregivers’ in-home context; scientific articles from 2017 to 2022. For the extraction and synthesis, two independent reviewers will quality analyze the inclusion and exclusion criteria and the data quality analysis. Disagreements will be resolved by a third reviewer.

## 1. Introduction

Population aging stems from the reduction in the mortality rate and birth rate, changing the age structure and leading to an epidemiological transition characterized by changes in health and disease patterns [1,2,3]. This reality is due to progress in public health, social and economic matters [3,4].

An older adult can be considered a person aged 65 or over [5]. Aging is a natural process resulting from biological, physical, psychological, and social changes that each person experiences uniquely [6,7]. The World Health Organization (WHO) [8] corroborates the uniqueness of older adults, stating that each person is unique in the sense that the development of the individual over time influences the ability and needs that characterize it.

A growing prevalence of chronic diseases (multimorbidity) is associated with aging, and it is important to consider lifelong care to achieve quality aging [9,10,11]. The WHO [8] defines chronic illness as a permanent, disabling illness that results from irreversible pathological changes, requiring special education of the person for rehabilitation or which may require long periods of care or supervision. According to Lefèvre et al. [12] and Palmer et al. [9], multimorbidity is defined by the coexistence of several disease conditions without any of them preceding the previous ones. This multimorbidity, according to several authors, can be grouped into cardiometabolic (Diabetes, obesity, hypertension), anxiety-depression, and related pain or back syndrome with radiating pain [13]. However, Silva [14] states that the higher prevalence of multimorbidity in the older adult population is not always associated with functioning dependence, aging is not necessarily associated with dependence and disability, and late identification of chronic diseases or their consequences can lead to loss of functioning. Functioning refers to the person’s capacity, at a given moment, to perform subsistence tasks, to get involved with the environment in which he is inserted, and to carry out his social participation [15].

Aging is a process that can be experienced with the greatest possible degree of autonomy for as long as possible [16] if the person is integrated into society to participate in it fully and actively.

The term “active aging” was implemented and defined as a process that takes place as a person ages and aims to promote opportunities in health, safety and participation so that quality of life can be promoted through the maintenance of functioning capacity (which results from the interaction between intrinsic (physical and mental) and extrinsic capabilities (an environment that involves the person)) [8].

The changes in the person associated with aging (such as cognitive impairments and decreased sensory acuity), together with multimorbidity and social isolation [14,17], which can coexist at this stage of life, tend to contribute to higher degrees of dependence [11], which can lead to more complex health and economic challenges [11,18].

The growing prevalence of multimorbidity will influence the quality of life of older adults [11], and priority should be given to the home context in a community care environment [18].

When there is a loss of functioning capacity, it will progress to greater dependence on other people, such as informal caregivers in-home context [17].

The informal caregiver can be the family member of the person being cared for, who lives with the person and does not receive remuneration for the care provided [19,20]. The informal caregiver takes care of the dependent person to facilitate their performance in daily life activities (personal hygiene, food, adherence to the therapeutic regimen, etc.) [17,19].

The task of caring for a dependent person is uninterrupted and may lead the caregiver to experience situations of stress and overload, in addition to the social and economic effects that will involve the entire life of the person being cared for [19]. It is essential to provide professional support to the caregivers in order to level physical and mental health [17,19].

In this context, the health of the person being cared for and the person who cares must be evaluated from a multi-level perspective.

The World Health Organization has promoted the paradigm shift in health care towards person-centered care, considering this strategy fundamental for the personalization of health care that ensures the real needs of people throughout the different stages of life [8].

Person-centered care ensures not only the human rights associated with ethics but also people’s satisfaction, therapeutic adherence, increased health outcomes, and quality of life [21,22,23]

Person-centered health intervention programs prioritize the person’s experience of illness, viewing the person cared for holistically by encompassing psychosocial factors [22]. These programs promote person empowerment in decision-making [24] and effectiveness in the person-healthcare professional relationship [22]

The institute of medicine defines person-centered care as being responsive to the preferences, values, and needs of the person, being respectful and responsive, and ensuring informed decision-making. It also states that this approach requires a true connection between the person and health professionals [25].

Older adults present distinct characteristics, for example, in the prevalence of multimorbidity that requires various supports from different areas, with the involvement of several professionals, which can promote fragmentation of care [26]. This age group presents distinct health, social, and economic needs that can be facilitated by the implementation of person-centered interventions.

Thus, person-centered health intervention programs require teamwork to know the person holistically for the development of individualized care plans that ensure that social and mental health needs are met, as well as other health care needs equally, and are a possible response to the unique characteristics of the older adults and their caregiver [25].

This review aims to determine which person-centered healthcare interventions exist for older adults (aged 65 and over) and their caregivers’ in-home context.

This review will be undertaken to answer the following question: Which person-centered healthcare interventions exist for older adults and their caregivers’ in-home context?

## 2. Materials and Methods

This protocol was performed according to the Preferred Reporting Items for Systematic Reviews and Meta-Analyses (PRISMA) and was registered in the International Prospective Register of Systematic Reviews (PROSPERO) under the registration number: CRD42022303687.

Randomized controlled trials will preferably be included.

This protocol was carried out in May 2022, and the systematic literature review is intended to be completed by the end of December 2022.

### 2.1. Eligibility Criteria

#### 2.1.1. Population

The inclusion criteria are studies about care person-centered interventions for older adults (aged 65 years or older) and their caregivers.

Studies in which participants do not have an informal caregiver will be excluded.

The summary of the inclusion and exclusion criteria is show in Table 1.

#### 2.1.2. Intervention

The review will include studies on person-centered healthcare interventions aimed at achieving health gains in older adults and their informal caregivers, within a community context in any geographic area.

The summary of the inclusion and exclusion criteria is show in Table 1.

#### 2.1.3. Comparison

This systematic literature review will preferably include studies with a comparative group.

The summary of the inclusion and exclusion criteria is show in Table 1.

#### 2.1.4. Primary Outcome

The primary outcome will be to verify which health person-centered care interventions for older adults and their caregivers exist.

The data will preferably be of a quantitative nature, such as averages, measures of prevalence or incidence, and frequencies. It is synthesized from quantitative studies.

The summary of the inclusion and exclusion criteria is show in Table 1.

#### 2.1.5. Study Design

This systematic literature review will preferably include quantitative studies.

The summary of the inclusion and exclusion criteria is show in Table 1.

#### 2.1.6. Context

Studies that include person-centered health care interventions in-home context will be included.

The summary of the inclusion and exclusion criteria is show in Table 1.

### 2.2. Search Strategy

#### 2.2.1. Data Sources

Bibliographic research and consultation of databases: PubMed, CINAHL, MedicLatina, Scopus, and MEDLINE.

#### 2.2.2. Search Terms

The research will include the combination of four key concepts according to Medical Subject Headings (MeSH) terms and using: (“older adults”, “elderly”, “aged”), (“caregivers”, “caregiver burden”), “patient-centered care” and the Boolean descriptors OR and AND, as follows: [(“older adults”) OR (elderly) OR (“aged”)] AND [(“caregivers”) OR (“caregiver burden”)] AND [“patient-centered care”].

The strategy will be adapted according to each database and will be restricted to the period from January 2017 to September 2022, in English, Portuguese and Spanish.

### 2.3. Data Collection and Analysis

#### 2.3.1. Selection of Studies

The studies obtained in the research through each database will be exported to EndNote X9, and duplicates will be removed.

To minimize bias, two reviewers will independently assess the inclusion of studies by reading the title, abstract, and keywords. Those that do not meet the inclusion criteria in this systematic literature review will be excluded. A third reviewer will be consulted in case of disagreements or doubts. Then the full texts will be evaluated. The selection process will be presented through a PRISMA flowchart with the screening results in the different phases.

#### 2.3.2. Data Extraction

Initially, in the data extraction phase, a descriptive evaluation of each study will be carried out using the extraction instrument according to the research question. Information will be exposed: study objective, participants, authors, interventions, results, and main conclusions.

Data extraction will be performed by the same two reviewers independently. If there is any disagreement, a third reviewer will assess.

#### 2.3.3. Quality Appraisal

The JBI quality assessment tool (https://jbi.global/critical-appraisal-tools (accessed on 20 December 2022)) will be used to assess published articles’ reliability, relevance, and results.

Two reviewers will independently assess the quality, and a third reviewer will make a new assessment to break the tie in case of disagreement.

The result of the quality assessment of each study will be presented. Therefore, all studies selected up to this stage will be included. In this way, it will be possible to perceive the quality of the evidence produced within the scope of this review. Low-quality studies will be excluded.

#### 2.3.4. Strategy for Data Synthesis

The synthesis and analysis of the results will be written and structured to answer the question presented for this Systematic Review of Literature. Healthcare interventions, study objectives, and target population will be presented.

The characteristics of the studies to be included must respond to the following data: person-centered care interventions focused on older adults and their caregivers, in-home context, assessment instruments used, and which results (health gains) came from the implementation of these interventions.

A data table will be constructed to summarize the answers to the research questions. These data will be grouped in a table with the characteristics of the included studies. The author, year, sample, gender, objectives, methods, interventions, results, conclusions, and a data compilation scheme will be made. This table aims to summarize the data collected and facilitate the analysis and discussion of the data collected.

Tables, graphs, and/or figures will be prepared to present the results of the synthesized data to facilitate the reader to compare the findings of each study included in the review.

All team members will participate in this process.

Involvement of the person in care and the public: There is no involvement of the person in care or the public in the conduct and development of this systematic literature review.

Ethics and disclosure: As only secondary data will be analyzed, ethical approval for this study is not required. This scientific article is a systematic literature review protocol. Results will be disseminated through peer-reviewed publications.

## 3. Discussion

The older adult population is unequivocally an extract of the population that lacks specialized health care at this stage of life.

Multimorbidity and the loss of functioning result in the aging process that may require care provided by others in-home context. Informal caregivers emerge with specifications to learn and self-manage the burden of caring for a dependent person.

All the factors described require a social and financial adaptation with a strong impact on the quality of life of older adults and their caregivers. These changes can be adapted to strategies that translate into effective health gains, such as quality of life.

Person-centered care is intended to provide an effective and personalized intervention to the person being cared for and may be the answer to complex issues involving aging, loss of functioning associated with aging, or resulting multimorbidity and their exacerbations. This intervention should not forget the person that takes care of the older adult but add their health and learning needs.

This systematic literature review aims to understand which healthcare interventions are most efficient through the recognition of health gains for older adults and their caregivers.

With the elaboration of this protocol, we intend to guarantee the process’s rigor, clarity, and quality. For this, we involved two reviewers in the multiple stages of identification and selection of studies in the databases.

With this review, we intend to contribute to practice (the effective implementation of health strategies and interventions that focus on the person being cared for and their actual needs) based on scientific evidence, with the future objective of further investigations contributing to this result.

Although person-centered care is widely seen as the answer to the complexity of the older adult, this review also aims to understand whether there are effective person-centered health programs and whether these programs bring benefits not only to the dependent older adults but also to their caregivers as a dyad rather than as separate individuals.

## Figures and Tables

**Table 1 jpm-13-00027-t001:** Summary of inclusion and exclusion criteria.

Inclusion Criteria	Exclusion Criteria
Older adults (aged 65 years or older) Older adults with multimorbidity Caregivers of older adults Implementation of a person-centered health intervention program with health gains Person-centered health intervention program in the home context Measuring health gains for the older adult and informal caregivers Articles from 2017 to 2022 Articles in English, Spanish or Portuguese. Articles with a comparative group.	Older adults without caregiver Implementation of a person-centered health intervention program without significant health gains Person-centered health intervention program only implemented and evaluated in a non-home setting Measuring health gains only for the older adult or only for the caregivers

## Data Availability

Not applicable.

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
