# Peer review of "Person-Centered Health Intervention Programs, Provided at Home to Older Adults with Multimorbidity and Their Caregivers: Protocol for a Systematic Review"

_jpm, 2022, doi:10.3390/jpm13010027_

Round 1

Reviewer 1 Report

I recommend to clariffy what is the person-centered health intervention program in the Introdution. I recommend to add more actual foreign titles, as well. There is incorrectly "heath" instead of health in the title. 

Author Response

Response to Reviewer 1 Comments

Point 1: I recommend to clariffy what is the person-centered health intervention program in the Introdution.

Response 1: As recommended we clarify what is a person-centered health intervention program:

“The World Health Organization has promoted the paradigm shift in health care towards person-centered care, considering this strategy fundamental for the personalization of health care that ensures the real needs of people throughout the different stages of life [8].

Person-centered care ensures not only the human rights associated with ethics, but also people's satisfaction, therapeutic adherence, increased health outcomes, and quality of life [21–23]

Person-centered health intervention programs prioritize the person's experience of illness, viewing the person cared for holistically by encompassing psychosocial factors [22]. These programs promote person empowerment in decision-making [24] and effectiveness in the person-healthcare professional relationship [22]

The institute of medicine defines person-centered care as care that is responsive to the preferences, values, and needs of the person, being respectful and responsive, and ensuring informed decision-making. It also states that this approach requires a true connection between the person and health professionals [25].

The older adult person presents distinct characteristics, for example, in the prevalence of multimorbidity that requires various supports from different areas, with the involvement of several professionals, which can promote fragmentation of care [26]. This age group presents distinct health, social, and economic needs that can be facilitated by the implementation of person-centered interventions.

Thus, person-centered health programs require teamwork to know the person holistically for the development of individualized care plans that ensure that social and mental health needs are met, as well as other health care needs equally, and are a possible response to the unique characteristics of the older adults and their caregiver [25].”

Point 2: I recommend to add more actual foreign titles, as well.

Response 2: As recommended we add the following relevant titles:

“Mackenbach, J. P. Omran’s ‘Epidemiologic Transition’ 50 years on. Int J Epidemiol 2022 vol. 51,no. 4, pp. 1054–1057. DOI: 10.1093/ije/dyac020.

 Mavritsakis, N.; Mîrza, C.M.; Tache, S. Changes related to aging and theories of aging. Health, Sports & Rehabilitation Medicine 2020, vol. 21, no. 4, pp. 252–255, DOI: 10.26659/pm3.2020.21.4.252.

Palmer, K. et al. Multimorbidity care model: Recommendations from the consensus meeting of the Joint Action on Chronic Diseases and Promoting Healthy Ageing across the Life Cycle (JA-CHRODIS) Health Policy (New York) 2018 vol. 122, no. 1, pp. 4–11, DOI: 10.1016/j.healthpol.2017.09.006.

Fabbri, E.; Zoli, M.; Gonzalez-Freire, M.; Salive, M.S.; Studenski, S.A.; Ferrucci, L. Aging and Multimorbidity: New Tasks, Priorities, and Frontiers for Integrated Gerontological and Clinical Research. J Am Med Dir Assoc 2015, vol. 16, no. 8, pp. 640–647, DOI: 10.1016/j.jamda.2015.03.013

Lindt, N.; van Berkel, J.; Mulder, B. C. Determinants of overburdening among informal carers: a systematic review, BMC Geriatr. 2020, vol. 20, no. 1, p. 304, DOI: 10.1186/s12877-020-01708-3.

Mead, N.; Bower, P. Patient-centred consultations and outcomes in primary care: a review of the literature. Patient Educ Couns 2022 vol. 48, no. 1, pp. 51–61, DOI: 10.1016/S0738-3991(02)00099-X.

Epstein, R. M.; Fiscella, K.; Lesser, C. S.;  Stange, K.S. Why The Nation Needs A Policy Push On Patient-Centered Health Care. Health Aff 2010, vol. 29, no. 8, pp. 1489–1495, DOI: 10.1377/hlthaff.2009.0888.

Barry, M. J.;  Edgman-Levitan, S. Shared Decision Making — The Pinnacle of Patient-Centered Care. New England Journal of Medicine 2012, vol. 366, no. 9, pp. 780–781, DOI: 10.1056/NEJMp1109283.

Byrne, K.; Frazee, K.; Sims-Gould, J.; Martin-Matthews, A. Valuing the Older Person in the Context of Delivery and Receipt of Home Support. Journal of Applied Gerontology 2012, vol. 31, no. 3, pp. 377–401, DOI: 10.1177/0733464810387578.”

Point 3: There is incorrectly "heath" instead of health in the title.

Response 3: As recommended the error has been rectified.

Thank you for the suggestions for improvement!

Vânia Nascimento

Prof. Dr. César Fonseca

Prof. Dr. Lara Guedes De Pinho

Prof. Dr.Manuel Lopes

Reviewer 2 Report

Thank you for giving me to review your manuscript. This manuscript is interesting and scientifically meaningful for considering person-centered health intervention programs provided at home to older adults with multimorbidity and their caregivers. Regarding the contents, the following revision should be considered.

In the whole manuscript, the authors should use not “elderly” but “older” in authentic articles. 

The background has too many paragraphs. The authors should focus on theory building, the problems, and the research question paragraphs. Different paragraphs contain mixed contents. The first paragraph should focus on general information regarding patient-centered care in international contexts. Moreover, the second and third paragraphs should introduce the research question as the theoretical and conceptual framework.

The introduction should include the international contexts and research questions of this study.

This study should describe why the review process used the suggested search engines to review comprehensively.

The method section should include a table with inclusion and exclusion criteria.

The discussion should describe more regarding this review's outstanding points.

There are multiple typos. The authors should revise the manuscript intensively.

Author Response

Response to Reviewer 2 Comments

Point 1: In the whole manuscript, the authors should use not “elderly” but “older” in authentic articles.

Response 1: As recommended “elderly” has been a substitute for “older” or “older adult”.

Point 2: The background has too many paragraphs. The authors should focus on theory building, the problems, and the research question paragraphs. Different paragraphs contain mixed contents. The first paragraph should focus on general information regarding patient-centered care in international contexts. Moreover, the second and third paragraphs should introduce the research question as the theoretical and conceptual framework.

Response 2: As recommended the background was changed:

“The WHO has been promoting the paradigm shift in health care towards person-centered care, considering this strategy as fundamental for the personalization of care, but, globally, the implementation of person-centered health intervention programs is still in an embryonic stage. Older adults have high morbidity rates which are often precursors to functional dependence on informal caregivers. Person-centered health intervention programs may be the answer to the vulnerability of older adults and their caregivers, but they are not yet intensively implemented. This systematic literature review aims to identify which person-centered health programs exist in the home setting for this population and show the health gains.”

Point 3: The introduction should include the international contexts and research questions of this study.

Response 3: As recommended the international context that meets the national one was introduced and articles that corroborate this context were cited, namely:

“Mackenbach, J. P. Omran’s ‘Epidemiologic Transition’ 50 years on. Int J Epidemiol, 2022 vol. 51, no. 4, pp. 1054–1057, DOI: 10.1093/ije/dyac020.

Palmer, K. et al. Multimorbidity care model: Recommendations from the consensus meeting of the Joint Action on Chronic Diseases and Promoting Healthy Ageing across the Life Cycle (JA-CHRODIS). Health Policy (New York) 2018, vol. 122, no. 1, pp. 4–11, DOI: 10.1016/j.healthpol.2017.09.006.

Lindt, N.; van Berkel, J.; Mulder, B. C. Determinants of overburdening among informal carers: a systematic review,” BMC Geriatr 2020, vol. 20, no. 1, p. 304, DOI: 10.1186/s12877-020-01708-3.

Epstein,R. M.; Fiscella,K.; Lesser,  C.S.; Stange, K.C. Why The Nation Needs A Policy Push On Patient-Centered Health Care. Health Aff 2010, vol. 29, no. 8, pp. 1489–1495, DOI: 10.1377/hlthaff.2009.0888.

Mead, N.; Bower, P. Patient-centred consultations and outcomes in primary care: a review of the literature. Patient Educ Couns 2002, vol. 48, no. 1, pp. 51–61, DOI: 10.1016/S0738-3991(02)00099-X.

Stewart, M. et al. The impact of patient-centered care on outcomes. J Fam Pract. 2000 vol. 49, no. 9, pp. 796–804.

Barry, M.J.; Edgman-Levitan, S. Shared Decision Making — The Pinnacle of Patient-Centered Care,” New England Journal of Medicine 2012, vol. 366, no. 9, pp. 780–781, DOI: 10.1056/NEJMp1109283.

Byrne, K.; Frazee, K.; Sims-Gould, J.; Martin-Matthews, A. Valuing the Older Person in the Context of Delivery and Receipt of Home Support. Journal of Applied Gerontology 2012, vol. 31, no. 3, pp. 377–401, DOI: 10.1177/0733464810387578.”

And research questions of this study: “This review will be undertaken to answer the following question: Which person-centered health care interventions exist for older adults and their caregivers in the home context?”

Point 4: This study should describe why the review process used the suggested search engines to review comprehensively.

Response 4: I would humbly ask reviewer 2 to clarify this issue so that we can respond clearly.

Point 5: The method section should include a table with inclusion and exclusion criteria.

Response 5: As recommended, a table was added with the inclusion and exclusion criteria of the articles:

2.1.7. Inclusion and Exclusion criteria

Table 1. Summary of inclusion and exclusion criteria

Inclusion criteria

Exclusion criteria

·         Older adults (aged 65 years or older)

·         Older adults with multimorbidity

·         Caregivers of older adults

·         Implementation of a person-centered health intervention program with health gains

·         Person-centered health intervention program in the home context

·         Measuring health gains for the older adult and informal caregivers

·         Articles from 2017 to 2022

·         Articles in English, Spanish or Portuguese.

·         Articles with a comparative group.

·         Older adults without caregiver

·         Implementation of a person-centered health intervention programme without significant health gains

·         Person-centered health intervention programme only implemented and evaluated in a non-home setting

·         Measuring health gains only for the older adult or only for the caregivers

Point 6: The discussion should describe more regarding this review's outstanding points.

Response 6: As recommended the most important point of this review was added: "Although person-centered care is widely seen as the answer to the complexity of the older adult, this review also aims to understand whether there are effective person-centered health programs and whether these programs bring benefits not only to the dependent older adults but also to their caregivers, as a dyad rather than as separate individuals".

Point 7: There are multiple typos. The authors should revise the manuscript intensively.

Response 7: As recommended the article has been revised, however, a new revision will be carried out by a native English translator after approval of the article.

Thank you for the suggestions for improvement!

Vânia Nascimento

Prof. Dr. César Fonseca

Prof. Dr. Lara Guedes De Pinho

Prof. Dr.Manuel Lopes

Round 2

Reviewer 2 Report

The manuscript has been considerably improved. I think that this paper is suited for inclusion in our journal.